# Effectiveness of Ethyl Formate as a Fumigant of *Blattella germanica* and *Periplaneta americana* (Blattodea: Ectobiidae, Blattidae) in Cross-Border Trade Transportation

**DOI:** 10.3390/insects12111010

**Published:** 2021-11-09

**Authors:** Hyun-Kyung Kim, Hanchan Park, Seung-Ju Seok, Yejin Kyung, Gil-Hah Kim

**Affiliations:** Department of Plant Medicine, College of Agriculture, Life and Environment Science, Chungbuk National University, Cheongju 28644, Korea; nshk0917@gmail.com (H.-K.K.); hanchan@sbcc.kr (H.P.); tmdwn405@naver.com (S.-J.S.); dpwls6308@naver.com (Y.K.)

**Keywords:** *Blattella germanica*, *Periplaneta americana*, cockroach, ethyl formate, fumigation

## Abstract

**Simple Summary:**

Fumigation is one effective cockroach control method. Ethyl formate (EF) has recently been employed as a grain fumigant and has been evaluated as relatively safer than other fumigants. In this experiment, the effects of an ethyl formate fumigant on two species of cockroaches were investigated. Cockroach nymphs and adults had 100% mortality, but the effect on egg hatching inhibition was weak. Therefore, ethyl formate could be used as a fumigant if the control period and its usage should be adjusted considering the developmental stage of the cockroach.

**Abstract:**

Cockroaches cause problems as pests not only locally in residential areas but also internationally, as they can disperse across borders in transport vessels. We investigated the effects of the ethyl formate (EF) fumigant on all developmental stages of *Blattella germanica* and *Periplaneta americana*. For *B. germanica* eggs, the hatching inhibition rate increased directly proportionately with the increasing treatment concentration of EF, but the 100% control was not observed. *P. americana* eggs did not show any fumigation effect, even after exposure to 60 mg/L of fumigant in a 12 L desiccator. Adults and nymphs of the two species showed various fumigation effects dependent on the concentration in the 12 L desiccator. When EF was applied at the lethal concentration for 99% mortality (LCT_99_) values of 35 mg/L for 4 h (78.5 mg·h/L) and 60 mg/L for 2 h (70.8 mg·h/L), respectively, adults and nymphs of both species had 100% mortality in a 0.65 m^3^ fumigation chamber with a 20% loading ratio. However, no significant difference from the control was observed in the egg stage of either species of cockroach. The results of this experiment indicate that EF can be used as a fumigant for cross-border transport vessels if the control period occurs during the cockroach developmental stage and continuous refumigation is performed.

## 1. Introduction

There are approximately 4600 species of cockroach in the world, but only a few species live with people [1,2]. The most common species in human habitats are the German cockroach (*Blattella germanica*), American cockroach (*Periplaneta americana*), Australian cockroach (*P. australasiae*), large brown cockroach (*P. brunnea* (Burmeister)) and oriental cockroach (*Blatta orientalis*), which predominate around the world [3,4,5].

These species are not only nuisances in residential areas but also transmit pathogens due to dietary habits and cause allergic disease through their feces [6,7,8,9,10,11]. In addition to human-dominated habitats, cockroaches can be problematic in containers, cargo ships and aircraft that travel between countries [12,13,14,15,16]. Chemical control in the form of baits, traps, foggers, residual applications and occasional fumigation is generally used to control cockroach populations [14,15,17,18,19]. In an enclosed space, such as a container or aircraft, fumigation is one of the methods by which pest populations can be effectively controlled in a short time [20]. Total release foggers (TRFs), as called bug bombs, are widely used to kill insects and contain aerosol propellants for fumigation. Although TRFs are often used to control pests in the indoor environments of residential areas, insecticidal baits are more effective, which suggests that the aerosol particles of TRFs may not reach the niches in which cockroaches reside. Furthermore, the residues created by TRFs are problematic [21,22,23].

Ethyl formate (EF) is a colorless gas that exists in nature. The effects of EF as a fumigant have already been reported for many quarantine pests of plants, such as grape mealybug (*Pseudococcus maritimus*), long-tailed mealybug (*P. longispinus*), western flower thrips (*Frankliniella occidentalis*), and onion thrips (*Thrips tabaci*) [24,25,26]. EF is effective after a short treatment time, and because it metabolizes quickly, it is easily decomposed, leaving little residue [27,28,29,30]. Although the effects of EF fumigation have been researched for many pests, studies on medical pests are insufficient.

In this study, the effects of EF fumigation on *B. germanica* and *P. americana* were investigated and the potential use of EF as a control agent in intercountry vehicles was assessed.

## 2. Materials and Methods

### 2.1. Insects

*B. germanica* and *P. americana* were maintained in the insect toxicology laboratory of Chungbuk National University for more than 3 years without exposure to any known insecticides. They were mass reared with dog food (Super Jindo, Daehan Feed. Co. Ltd., Incheon, Korea) in plastic cages (54 × 40 × 27 cm) and were incubated at 28 ± 2 °C and 60 ± 10% relative humidity (R.H.) under 16:8 h light:dark (L:D) conditions.

### 2.2. Fumigant

Ethyl formate (C_3_H_6_O_2_, 99.6%) as a colorless liquid, was supplied by Fuoviders Inc. (Asan, Korea).

### 2.3. Fumigation Experiments

Fumigation experiments involving EF and the two cockroach species were conducted using a 12 L desiccator (Duran, Mainz, Germany) and a 0.65 m^3^ fumigation chamber modified according to Cho et al. (2020). Ten cockroaches of each adult and nymph developmental stage were placed in a breeding dish (9 × 5 cm) containing food; the dish was placed in a 12 L desiccator, which was sealed with vacuum grease (high-vacuum grease, Dow Corning, Midland, TX, USA). Adults and nymphs were inoculated without distinction between males and females. EF was placed on filter paper (90 mm I.D.) in the desiccator using a 100 µL gastight syringe (Hamilton, NV, USA). Various amounts (10–70 mg/L) of the fumigant were applied, and the experiments were conducted at 20 °C for exposure periods of 2 h and 4 h. Adults and nymphal mortality were observed 4, 8, 12, 24, 48, 72 and 96 h after treatment.

Each breeding dish containing cockroaches was placed in a 0.65 m^3^ fumigation chamber (230 × 50 × 50 cm) with timber at a loading ratio of 20%. The breeding dishes were placed at the top, middle and bottom of the fumigation chamber. The EF concentrations were 35 mg/L and 60 mg/L, which the treatment concentrations were selected by statistically estimating the 99% lethal concentration and time value (LCT_99_) of the adult and nymph stages, with 2 h and 4 h of exposure at 20 °C, respectively, in the 12 L desiccator experiment.

The hatching rate (%) was determined using the number of hatched egg sacs (ootheca). The hatching inhibition rate (%) was determined by inoculating 5 egg sacs in each breeding dish until 10 weeks after treatment and was calculated as [(number of hatched nymphs in the control–number of hatched nymphs in the treatment)/number of hatched nymphs in the control] × 100. The experimental conditions were the same as those in the adult and nymphal fumigation test (i.e., a 12 L desiccator and 0.65 m^3^ fumigation chamber were used). The fumigation treatment was conducted at 20 °C with 4 h exposure.

The control was not treated with any fumigant. All the experiments were repeated 3 times, and the dishes were incubated after treatment at 28 ± 2 °C and 60 ± 10% R.H. with a 16:8 h (L:D) photoperiod.

### 2.4. Gas Concentration and Sorption Measurement

From the 12 L desiccators, 50 mL of EF gas was collected in Tedlar gas sampling bags (1 L, SKC, Dorset, UK) using a syringe (100 mL, Hamilton, NV, USA). The concentration and time (CT) values were calculated by collecting gases at 30 min and 1, 2 and 4 h after EF treatment (AFHB/ACIA, 1989). The concentrations were analyzed using gas chromatography (Agilent Technology 6890N, FID, Santa Clara, CA, USA) with the GC conditions described in Kyung et al. (2019). Briefly, the injector temperature of the flame ionization detector (FID) was 200 °C, the detector temperature was 240 °C, the oven temperature was 100 °C, and an HP-5 column was utilized (0.32 mm × 30 m, Agilent Technology, Santa Clara, CA, USA).

The sorption rate of EF was determined using a 12 L desiccator with 0%, 2% and 20% timber loading ratios (*w*/*v*). EF was applied at concentrations of 35 and 60 mg/L, and the experiments were conducted at 20 °C for 4 h and 2 h, respectively. The gas concentrations for sorption were determined at 30 min, 1 h, 2 h and 4 h after treatment. *C*/*C*_0_ values were calculated as the concentration at each time point after treatment (*C*) divided by the initial concentration after treatment (*C*_0_). There were no timber blocks in the 12 L desiccators in the control treatment.

### 2.5. Statistical Analysis

The hatching inhibition rate and mortality of *B. germanica* and *P. americana* eggs due to EF fumigation were compared and analyzed using Tukey’s test (SAS Institute, 2009). The LCTs of EF for *B. germanica* and *P. americana* adults and nymphs were calculated using probit analysis [31].

## 3. Results

### 3.1. Inhibitory Effects of EF on Hatching

The inhibitory activity of EF on the hatching of egg sacs from two species of cockroaches was investigated in a 12 L desiccator (Table 1).

The hatching inhibition of the eggs from the two cockroach species revealed that fumigation with EF does not yield 100% mortality. The hatching rate of *B. germanica* egg sacs decreased with increasing treatment concentration. However, the hatching rate of *P. americana* egg sacs was 86.7%, even at 60 mg/L, which was not significantly different from that in the control. The hatching inhibition rate according to the number of nymphs was 96.8% for *B. germanica* and 13.9% for *P. americana* in the 60 mg/L treatment. Even the 60 mg/L EF treatment in the 12 L desiccator did not result in 100% control of the eggs from either cockroach species.

The effects of EF fumigation on eggs from 2 cockroach species were observed in a 0.65 m^3^ fumigation chamber (Table 2). A 100% control effect was not observed for the eggs of either species when treated with the LCT_99_ values (78.5 mg·h/L) associated with the adult and nymphal cockroaches. There was also no significant difference in the effect of fumigation according to the location of the fumigation chamber.

### 3.2. Effects of EF Fumigation on Adults and Nymphs

The effects of fumigation were investigated in adults and nymphs of two cockroach species (Figure 1).

The mortalities of adult and nymphal *B. germanica* were observed after 2 and 4 h of EF exposure (Figure 1A). In the 18 mg/L EF treatment group, the mortality of *B. germanica* adults and nymphs decreased after 8 h and 2 h of EF exposure, respectively. Then, over time, the mortality increased again. However, the mortality of treated adults after 4 h was not significantly different from those immediately after treatment or up to 96 h after treatment, while the mortality of *B. germanica* nymphs decreased in the 12 mg/L treatment. Both developmental stages were treated with 15 mg/L for 4 h and showed 100% mortality.

Mortality of *P. americana* adults and nymphs was observed according to exposure time and EF concentration (Figure 1B). EF concentrations of more than 60 mg/L were needed to yield 100% mortality of *P. americana* adults and nymphs 24 h after treatment. Adults (30 mg/L) and nymphs (35 mg/L) showed 100% mortality after 4 h of EF exposure, and as the fumigant concentration decreased, the effects of EF on the nymphs decreased, although the symptoms appeared similar to those achieved by knockdown.

The effects of EF fumigation on the adults and nymphs of two species of cockroach were investigated in a 0.65 m^3^ fumigation chamber containing timber blocks with a 20% loading ratio (*w*/*v*) (Table 3). In the adults and nymphs of both *B. germanica* and *P. americana*, the LCT_99_ (78.5 mg·h/L) for EF for an exposure period of 4 h led to 100% mortality after 24 h. In addition, EF fumigation was 100% effective against all the insects, even given exposure to the LCT_99_ (70.8 mg·h/L) for 2 h. Regardless of the spatial location of the fumigation chamber, EF fumigation was 100% effective against both cockroach species.

### 3.3. Timber Sorption Rate of EF

The sorption rates of timber blocks exposed to EF for 2 h and 4 h were analyzed (Figure 2).

When timber blocks were treated with 35 mg/L of EF at loading ratios of 2% and 20%, the concentration (*C*/*C*_0_) decreased by 0.9 and 0.88, respectively, after 4 h of exposure. When an EF concentration of 60 mg/L and loading ratios of 2% and 20% were used, the concentrations were 0.94 and 0.85, respectively, after 2 h of exposure. However, in the high-dose treatment, a high sorption rate was observed in the timber with a high loading ratio (20%). The EF concentrations in the control (without timber) did not show a significant reduction according to the treatment doses.

## 4. Discussion

Cockroaches are accidentally transported across geographic boundaries, and they cause damage not only in residential spaces but also contribute to the destruction of goods and threaten human health. Cockroach control is essential not only for housing but also for galleys and mess rooms for crew and passengers on ships or aircraft [15,16,19]. The use of fumigants is one of the alternative approaches for controlling their populations. Various fumigants are used in quarantine settings, and some have disadvantages: phosphine (PH_3_) damages copper, sulfuryl fluoride (SF) has comparatively low effectiveness at the egg stage, and chlorine dioxide is not suitable for plastics [20,32,33,34,35]. In the study on cockroach control using atmospheric conditions, treatment of *B. orientalis* and *P. americana* with 60% CO_2_ had small effects on ootheca, resulting in the need for longer treatment times [36,37]. In addition, the control of *B. germanica* and *P. americana* through hypoxia mediated by nitrogen gas was studied, but this approach is difficult to apply due to the risk of explosion [38]. For pest control in stores of dried fruits and cereals, EF has been used; this fumigant leaves almost no residue and has a shorter treatment time than other fumigants [24,27,29,30]. The characteristics of EF may make it suitable for cockroach control.

The effectiveness of fumigants will depend on the developmental stage of the insects [26]. The eggs and pupae of *B. orientalis* and *P. americana* were more tolerant than other developmental stages to a CO_2_ treatment; those of *F. occidentalis*, *Phthorimaea operculella*, *Carposina niponensis* and *Rhynchophorus ferrugineus* to a PH_3_ treatment; and those of *Tetranychus urticae* to an EF treatment [36,37,39,40,41,42,43]. It was also observed in this study that the EF treatment had a very small inhibitory effect on oothecal hatching. Since the eggs of cockroaches are protected by egg cases, cockroach control using fumigants should consider the hatching period [44]. The incubation period of eggs is 17.2 days for *B. germanica* and 34 days for *P. americana*, and the nymphal development periods are 40 and 165 days, respectively [45]. EF has a small effect on the eggs of both species of cockroach, but it is effective as a fumigant against nymphs and adults; therefore, effective control is possible if refumgation is performed considering the hatching time and developmental period of nymphs. When the EF exposure time was increased, the effectiveness was higher, even at a low concentration. However, decreased mortality of adults and nymphs was observed at low concentrations. In particular, the mortality of nymphs decreased compared with that of adults over time when the two groups were treated with a concentration below a certain level. Although EF treatment showed some effectiveness against the nymphs and adults of the two species of cockroaches studied, 100% control required treatment at a certain concentration or higher. The EF concentration needed for 100% control is thought to have a knockdown effect, indicating that this treatment concentration is very important for 100% control of cockroaches. Cockroach control using EF will be effective when specific doses are applied through a continuous control strategy, with refumgation being performed after a certain period of time in consideration of cockroach life cycles. Therefore, the results of this study might be helpful because they demonstrate that EF may be one of the fumigants that can be used as a cockroach control agent in transportation settings.

## 5. Conclusions

Our study on the fumigation effects of EF on *B. germanica* and *P. americana* found that adults and nymphs were highly susceptible to EF; however, the egg stage in both cockroaches had a very low hatching inhibition rate. Thus, ethyl formate can be used as a fumigant for cross-border transport vessels if continuous refumigation considering developmental stages is performed.

## Figures and Tables

**Figure 1 insects-12-01010-f001:**
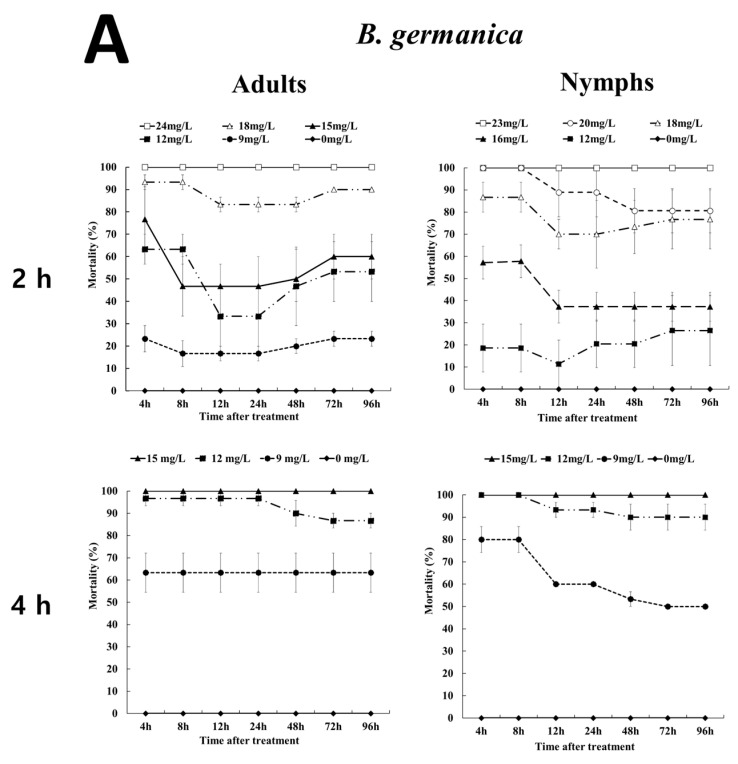
Fumigation effects of EF on *B. germanica* (**A**) and *P. americana* (**B**) adults and nymphs in a 12 L desiccator according to concentration and exposure time.

**Figure 2 insects-12-01010-f002:**
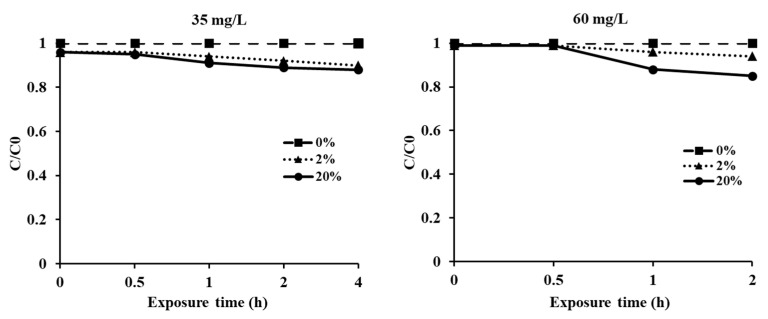
Concentrations of EF given different timber loading ratios (0, 2 and 20%) during fumigation in a 12 L desiccator with 35 mg/L and 60 mg/L for 4 h and 2 h, respectively.

**Table 1 insects-12-01010-t001:** Inhibitory effects of ethyl formate on the hatching rates of *B. germanica* and *P. americana* egg sacs for 10 weeks after a 4 h exposure in a 12 L desiccator.

Species	Concentration (mg/L)	CT (mg·h/L)	*n* ^a^	Hatched Ootheca (%, Mean ± SE) ^b^	No. of Hatched Nymphs	Hatching Inhibition Rate (%, Mean ± SE)
*B. germanica*	70	219.6	15	6.7 ± 6.7 c	13	97.4 ± 2.6 bc
	60	178.9	15	6.7 ± 6.7 c	16	96.8 ± 3.2 c
	50	137.5	15	46.7 ± 6.7 b	154	68.5 ± 4.4 bc
	40	115.2	15	53.3 ± 13.3 b	174	64.2 ± 9.4 abc
	30	77.4	15	73.3 ± 6.7 ab	265	46.0 ± 8.8 ab
	20	50.0	15	80.0 ± 11.5 ab	355	27.9 ± 13.1 a
	10	27.5	15	73.3 ± 6.7 ab	313	36.2 ± 5.7 a
	Control		15	100.0 ± 0.0 a	492	-
*P. americana*	60	178.9	15	86.7 ± 6.7 a	137	13.9 ± 11.7 a
	40	115.2	15	100.0 ± 0.0 a	151	6.6 ± 1.2 a
	30	77.4	15	80.0 ± 11.5 a	134	17.8 ± 12.0 a
	10	27.5	15	86.7 ± 6.7 a	128	20.4 ± 7.6 a
	Control		15	100.0 ± 0.0 a	162	-

^a^ Total number of *B. germanica* and *P. americana* egg sac (ootheca) tested. ^b^ The means within each cockroach followed by the same letters are not significantly different at *p* = 0.05 (Tukey’s test).

**Table 2 insects-12-01010-t002:** Inhibitory effects of ethyl formate on the hatching rates of *B. germanica* and *P. americana* egg sacs for 10 weeks after a 4 h exposure in a 0.65 m^3^ fumigation chamber containing a timber block with a loading ratio of 20%.

Concentration (mg/L)	Species	*n* ^a^	Location	Hatched Ootheca (%, Mean ± SE) ^b^	No. of Hatched Nymphs	Hatching Inhibition Rate (%, Mean ± SE)	CT (mg·h/L)
35	*B. germanica*	15	Top	80.0 ± 11.5 a	287	34.7 ± 5.6 a	78.5
		15	Middle	66.7 ± 6.7 a	272	37.6 ± 10.2 a	
		15	Bottom	73.3 ± 6.7 a	261	40.7 ± 6.2 a	
		15	Control	93.3 ± 6.7 a	445	-	
	*P. americana*	15	Top	80.0 ± 11.5 a	131	11.1 ± 4.8 a	
		15	Middle	73.3 ± 6.7 a	113	22.2 ± 6.7 a	
		15	Bottom	80.0 ± 0.0 a	139	4.4 ± 8.8 a	
		15	Control	86.7 ± 6.7 a	148	-	

^a^ Total number of *B. germanica* and *P. americana* egg sac (ootheca) tested; ^b^ The means within each cockroach followed by the same letters are not significantly different at *p* = 0.05 (Tukey’s test).

**Table 3 insects-12-01010-t003:** Fumigant toxicity of ethyl formate against *B. germanica* and *P. americana* nymphs and adults in a 0.65 m^3^ fumigation chamber 2 h and 4 h exposure containing timber block with a loading ratio of 20%.

Exposure Time	Concentration (mg/L)	Species	Stage	*n*	Location	Mortality	CT (mg·h/L)
4 h	35	*B. germanica*	Adult	30	Top	100.0 ± 0.0 a	78.5
				30	Middle	100.0 ± 0.0 a	
				30	Bottom	100.0 ± 0.0 a	
				30	Control	6.7 ± 3.3 b	
			Nymph	30	Top	100.0 ± 0.0 a	
				30	Middle	100.0 ± 0.0 a	
				30	Bottom	100.0 ± 0.0 a	
				30	Control	6.7 ± 6.7 b	
		*P. americana*	Adult	30	Top	100.0 ± 0.0 a	
				30	Middle	100.0 ± 0.0 a	
				30	Bottom	100.0 ± 0.0 a	
				30	Control	6.7 ± 3.3 b	
			Nymph	30	Top	100.0 ± 0.0 a	
				30	Middle	100.0 ± 0.0 a	
				30	Bottom	100.0 ± 0.0 a	
				30	Control	6.7 ± 3.3 b	
2 h	60	*B. germanica*	Adult	30	Top	100.0 ± 0.0 a	70.8
				30	Middle	100.0 ± 0.0 a	
				30	Bottom	100.0 ± 0.0 a	
				30	Control	6.7 ± 3.3 b	
			Nymph	30	Top	100.0 ± 0.0 a	
				30	Middle	100.0 ± 0.0 a	
				30	Bottom	100.0 ± 0.0 a	
				30	Control	6.7 ± 3.3 b	
		*P. americana*	Adult	30	Top	100.0 ± 0.0 a	
				30	Middle	100.0 ± 0.0 a	
				30	Bottom	100.0 ± 0.0 a	
				30	Control	6.7 ± 3.3 b	
			Nymph	30	Top	100.0 ± 0.0 a	
				30	Middle	100.0 ± 0.0 a	
				30	Bottom	100.0 ± 0.0 a	
				30	Control	6.7 ± 3.3 b	

Mortality (%, mean ± SE), followed by the different letters within columns, was significantly different at *p* < 0.05 by Tukey’s test (SAS Institute 2009).

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
