# Peer review of "Effectiveness of Ethyl Formate as a Fumigant of Blattella germanica and Periplaneta americana (Blattodea: Ectobiidae, Blattidae) in Cross-Border Trade Transportation"

_insects, 2021, doi:10.3390/insects12111010_

Round 1
Reviewer 1 Report
The research paper “Effectiveness of ethyl formate as a fumigant of Blattella germanica and Periplaneta americana (Blattodea: Ectobiidae, Blattidae) in cross-border trade transportation” represents an original, relevant, and interesting study on the problem of pest control methods. It addresses the revealing of the effects ethyl formate on two species of cockroaches. The paper is nicely written, the literature review in the introduction part is enough summarized, however, some issues indicated below should be addressed. The statistical analyses are very simple, although descriptive. But I didn’t find any explanation about the normality in the variable’s distributions.
I couldn’t provide revision for the further part of the manuscript, because line numbers were missing.
line 9: has been recently employed
line 10: “Fumigation is an effective cockroach control method” this sentence should be first in the simple summary.
lines13-14: “…is adjusted considering the 13 developmental stages of the cockroach”, this part of the sentence should be corrected to “ and its usage should be adjusted considering 13 developmental stages of the cockroach”
line 15: In the scientific texts, it is suggested to avoid the passive voice. For this reason, I suggest rephrasing the sentence: “Cockroaches cause problems as pests not only locally – in residential areas, but also internationally, as they can disperse across borders in the transport vessels”.
line 16: of the ethyl formate
lines 16-17: Here we investigated the effects of.. (avoid passive voice where it is possible)
line 19: the 100% control (but was the procent observed?)
line 29: the terms in the keyword section should not repeat the title, compliment but not repeat.
lines 32-33: Passive voice is not needed here “but only a few species live with people”!
line 34: Blattella germanica: the font size differs from the other text
line 45: can you add some text to describe the method (TRFs) to non-specialists?
lines 50-51: common names of these insects are missing, especially you’ve mentioned those for the cockroaches before
line 62: which food?
lines 65-66: I would suggest adding a more detailed description of the fumigant: the class of chemical compounds, physical and chemical properties (in brief), area of use
line 70: which were developmental stages?
line 75: which amounts of fumigant?
lines 81-83: not clear, please rephrase
lines 113-114: what do you mean by “data not shown”? the data should be present if your paper is based on those
Table 1: where is the explanation for “c” mark in the table?
Why are the line numbers not presented here? I can’t provide revision for the text after Table 1
Author Response
Thank you very much for your advice and kindness.

Reviewer 2 Report
Dear Authors,
The paper presents influence of the ethyl formate on mortality two species of Cockroach. The laboratory experiment methods are clearly discribed and obtained results showed mortality of the nymph and adults in rigorous experimental parameters. The assumptions of the experiment seems be correct, nevertheless are very narrow to talk about the proper effectiveness of this chemical compound. Authors suggested that the using ethyl formate as fumigant can be used against cockroaches populations, however, they are not explained whether the use of ethyl formate in a more natural setting than in a laboratory (e.g. slightly different temperature or humidity) will be effective. The utility of these results is low.
Author Response

(The authors gave the same response as above.)

Reviewer 3 Report
The manuscript provides a well conceived and designed study about the efficacy of the use of Ethyl Formate for the control of two cockroach species. It provides very clear data that can be used to make recommendations for management strategies. Only a handful of minor comments concerning presentation are made below.
Introduction: Last sentence is awkward. Perhaps change the word "possibility" to "potential use" and the word "examined" to "assessed". The original wording seems a bit too precise for the context.
Methods: Why were the temperatures changed from 20 to 28 C for the different parts of the experiment? It appears treatments were always at the lower temperature and incubation at the higher one. A sentence or two should be added to explain. Would this impact management strategies? If so expand in Discussion.In Table 3 it would be preferable to use the term "block location" rather than "locate" Font size needs to be increased throughout Figure 1. It would be preferable to place part A above part b, rather than side by side to facilitate this change.
Author Response

(The authors gave the same response as above.)

Round 2
Reviewer 1 Report
Dear authors,
the changes made are substantial and good.
Line 213: Please remove” Which can be”, cockroaches, accidentally transported…
line 2014: Try to avoid elaborated constructions “not only …but” Cause damage to residential … and contribute… try to keep structures simple
line 2018: the egg stage;
line 2018: “some fumigants” which ones? current formulation is very vague
line 2019: in the study
line 222: was studies or tested, but not researched
line 225-227: this sentence should be place at the very beginning of the paragraph, otherwise it disrupts the logic here
line 228: depends … and the reference is needed at the end of this sentence
line 254: it is better to use the whole title but not the abbreviation (EF)
Author Response
Reviewer 1
Thank you very much for your advice and kindness.
Line 213: Please remove” Which can be”, cockroaches, accidentally transported…
--> revised
line 2014: Try to avoid elaborated constructions “not only …but” Cause damage to residential … and contribute… try to keep structures simple
--> revised
line 2018: the egg stage;
--> It has not been modified as it is connected to the following sentence.
line 2018: “some fumigants” which ones? current formulation is very vague
--> revised to chlorine dioxide.
line 2019: in the study
--> revised
line 222: was studies or tested, but not researched
--> revised to studied
line 225-227: this sentence should be place at the very beginning of the paragraph, otherwise it disrupts the logic here
--> moved to the front of discussion.
line 228: depends … and the reference is needed at the end of this sentence
--> inserted
line 254: it is better to use the whole title but not the abbreviation (EF)
--> revised
Reviewer 2 Report
Thank you very much for the authors' reply, and I agree with their explanation. The authors pointed and supported answers based on the other papers data that the moisture does not significantly affect ethyl formate. The pointed paper (J. Econ. Entomol. (2019) 112, 2149-2156) should be cited and added to the manuscript because it is lacking.
Author Response
Reviewer 2
Thank you very much for your advice and kindness.
Thank you very much for the authors' reply, and I agree with their explanation. The authors pointed and supported answers based on the other papers data that the moisture does not significantly affect ethyl formate. The pointed paper (J. Econ. Entomol. (2019) 112, 2149-2156) should be cited and added to the manuscript because it is lacking.
--> Actually, similar papers have already been inserted as references.